# Projection of Air Pollution in Northern China in the Two RCPs Scenarios

Chengrong Dou [1], Zhenming Ji [1,2,3,*], Yukun Xiao [1], Zhiyuan Hu [1,2,3], Xian Zhu [1,2,3] and Wenjie Dong [1,2,3]

1   School of Atmospheric Sciences, Sun Yat-sen University, Zhuhai 519082, China;
    douchr@mail2.sysu.edu.cn (C.D.); xiaoyk5@mail2.sysu.edu.cn (Y.X.); huzhiyuan@mail.sysu.edu.cn (Z.H.);
    zhux53@mail.sysu.edu.cn (X.Z.); dongwj3@mail.sysu.edu.cn (W.D.)
2   Key Laboratory of Tropical Atmosphere-Ocean System, Ministry of Education, Zhuhai 519082, China
3   Southern Marine Science and Engineering Guangdong Laboratory, Zhuhai 519082, China
*   Correspondence: jizhm3@mail.sysu.edu.cn

**Abstract:** Air pollution in North China (NC) is an important issue affecting the economy and health. In this study, we used a regional climate model, the Weather Research and Forecasting Model with Chemistry (WRF-Chem) to project air pollution in NC and investigate the variations of air pollutions response to future climate changes, which probably has an implication to strategy and control policy for air quality in NC. A comprehensive model evaluation was conducted to verify the simulated aerosol optical depth (AOD) based on MODIS and MISR datasets, and the model also showed reasonable results in aerosol concentrations. Future changes of air pollution in the middle of the 21st century (2031–2050) were projected in the two Representative Concentration Pathways (RCP4.5 and RCP8.5) and compared with the situation in the historical period (1986–2005). In the two RCPs, the simulated averaged $PM_{2.5}$ concentration was projected with the highest values of 50–250 $\mu g \cdot m^{-3}$ over the Bohai Rim Economic Circle (BREC) in winter. The maximum AOD is in the Beijing–Tianjin–Hebei (BTH) region in summer, with an average value of 0.68. In winter, in the RCP4.5 scenario, $PM_{2.5}$ concentration and AOD obviously declined in BTH and Shandong province. However, in the RCP8.5 scenario, $PM_{2.5}$ concentration and AOD increased. Results indicated that air pollution would be reduced in winter if society developed in the low emission pathway. Precipitation was projected to increase both in the two RCPs scenarios in spring, summer, and winter, but it was projected to decrease in autumn. The planetary boundary layer height decreased in the two RCPs scenarios in the central region of NC in the summer and winter. The results indicated that changes of meteorological conditions have great impact on air pollution in future scenarios.

**Keywords:** air pollution; Representative Concentration Pathways (RCP) scenarios; projection; North China; meteorological influence



## 1. Introduction

With the development of the economy and urbanization, China has been suffering from an air pollution problem in the past two decades. Many previous studies revealed that in China, a significant decline in air quality has become more serious [1,2]. There was no doubt that air quality has been an important issue around the world. Urban air pollution is associated with increased mortality and morbidity in both developed and developing countries [3,4]. In 2019, the World Health Organization (WHO, 2019) [5] listed air pollution as the most important environmental problem that threatens public health and causes about 7 million deaths worldwide every year. Therefore, the assessment of air quality is a very critical issue. Air pollution in China is worse than other countries in East Asia, which has caused a severe impact on climate change [6–8]. A large number of previous studies have demonstrated that the effects of aerosol pollution on climate change were significant [9,10].

The further urbanization was a consequence of the growing industry. In the period of 1978–2017, more than 550 million migrants moved to cities, resulting in a large rising of urban population from 18 to 57% [11]. Figure 1 showed the population density of China in 2000. Most of the population was concentrated in the NC plain, the Yangtze River Delta, and the Pearl River Delta due the development of prosperous cities. The Beijing–Tianjin–Hebei (BTH) Economic Circle and the Bohai Rim Economic Circle (BREC) are located in NC. Therefore, the manufacturing and services industries in coastal areas are relatively developed.

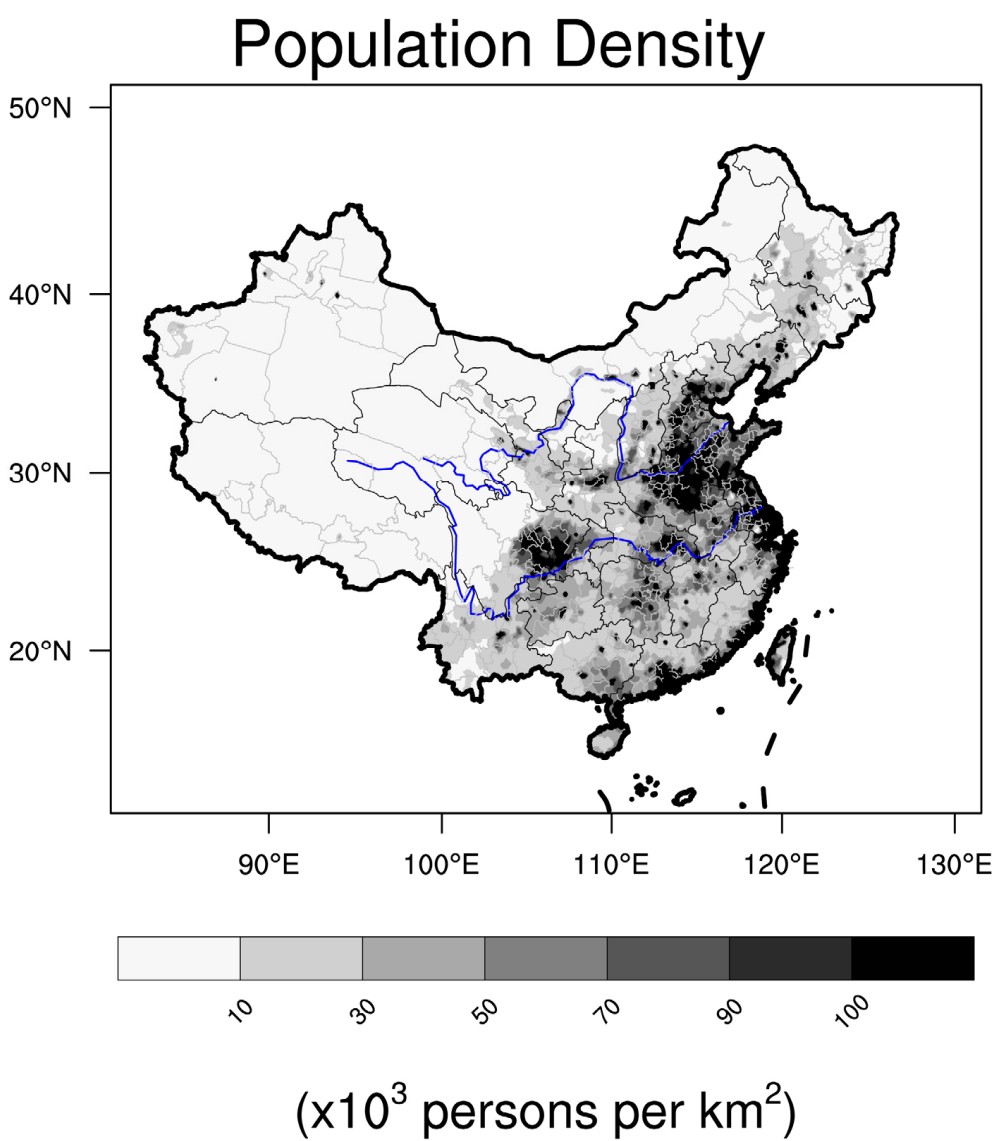

**Figure 1.** The density of population in China.

At present, air pollution has gradually shifted from the traditional total suspended particulate matter and inhalable particle matter (PM$_{10}$) and SO$_2$ pollution to a compound pollution that is a mixture of fine particles (PM$_{2.5}$) and polluting gases such as O$_3$, SO$_2$, and NOx in China. The fine particulate pollution threatens the health of more than 400 million people in NC. Haze was affected by both atmospheric aerosol loadings and weather conditions [12]. Recent studies have shown that the region suffers the highest particle matter (PM) concentrations and the longest pollution episodes [13]. The highest PM concentrations were observed, and more dust aerosols were observed in NC [14]. The annual population-weighted-average (PWA) concentrations of various pollutants

in NC were greater than those in Southern China, and PM$_{2.5}$ pollution in winter was worsening [15]. Many studies have explored the changes in air quality in different regions in the climate change scenarios. Yahya et al. (2017) used WRF-Chem to project the climate and air quality, and the results illustrated that the difference in the spatial distribution of meteorological elements were induced by variations of aerosols' emissions in future scenarios [16]. Another study based on RAMS-CMAQ projected changes of aerosol species between three RCP scenarios and indicated that the impact of climate change on different species tends to be consistent [17]. The study by Cai et al. (2013) simulated spatial distributions of PM$_{10}$ by WRF-Chem and manifested that northwestern and northern China are the two regions with the largest dust aerosol sources in East Asia [18].

Meteorological condition was another factor inducing the serious air pollution in NC, which influenced the processes of removal, diffusion, and transformation [19,20]. Climate change in NC might have caused the frequency, intensity, and duration of atmospheric compound pollution events to increase [21]. Several air pollution events were caused by weather conditions rather than sudden increases of emissions. He et al. (2013) revealed that local meteorological conditions were the main factors causing the day-to-day variations of SO$_2$, NO$_2$, and PM$_{10}$ for the winter of 2002–2007 in Lanzhou, Northwestern China [22]. In recent years, several extreme air pollutions in the vast areas of NC in winter were related to extreme weather conditions [23]. Furthermore, some studies had attempted to identify the correlation between pollutants and meteorological factors [24,25]. The results have shown that the low planetary boundary layer height (PBLH) [26], the weakening of northerly winds [27], the decrease of relative humidity [28], and the increase of sea level temperature [29] have also led to the increase of winter haze in eastern China.

Although there were a few studies focused on the projection of air pollutions, which was also based on the model simulation, few of them paid attention to those variations in NC. In this study, we focused on the projection of air pollutions in NC in the two RCP scenarios based on a high-resolution regional climate model WRF-Chem V.3.9.1. The main purpose was to investigate the variations of air pollutions response to future climate changes, which probably has an implication to strategy and control policy for air quality in NC.

## 2. Model, Data, and Methodology

### 2.1. Model and Simulation

2.1.1. WRF-Chem Model Configuration

WRF-Chem is a regional atmospheric chemistry model that fully couples a meteorological module and chemical mode online. Different from the global climate model (GCM) with coarse resolution, WRF-Chem can be used to simulate the feedback process on a wide range of spatial scales, as it is a non-hydrostatic model with domain nesting [30]. More details about WRF-Chem were described in a previous study [31]. Now, the development of WRF-Chem is more comprehensive, and the application is more extensive.

The 3.9.1 version of WRF-Chem (Figure S1) was used in this study, the projection was centered at 33°N, 103.3°E with 170 and 124 grid points in the east–west and north–south directions, respectively. The horizontal resolution was 27 km, and the vertical grid in the model consists of 27 levels ranging. The Lin microphysics scheme [32] was used in this simulation. The Rapid Radiative Transfer Model (RRTM) was applied to calculate long wave radiation processes [33]. Aerosol and gas phase chemistry were described, using the second Regional Acid Deposition Mode (RADM2) photochemical mechanism [34] and the Modal Aerosol Dynamics model for Europe (MADE), which incorporates the Secondary Organic Aerosol Model (SORGAM) [35,36]. RADM2 included 63 prognostic species and 136 reactions [37]. Aerosol direct feedback was turned on. Short-wave radiation was the Dudhia scheme [30]. More details about WRF-Chem simulations are summarized in Table 1.

**Table 1.** Physical and chemical scheme adopted in the WRF-Chem simulations.

| Model Configurations | Aerosol-Containing Feedback Mechanism |
|---|---|
| Microphysics scheme | Lin |
| Shortwave radiation scheme | Dudhia |
| Long wave radiation scheme | RRTM |
| Cumulus parameterization scheme | Multi-scale KF |
| Photolysis scheme | Fast-J |
| Gas chemical scheme | RADM2 |
| Aerosol chemistry scheme | MADE/SORGAM |
| Aerosol effect feedback | On |

2.1.2. Simulation Design

In this study, WRF-Chem simulated air quality and climate over NC for a historic period (1986–2005) and future decades (2031–2050) in the RCP4.5 and RCP8.5 scenarios. The initial and boundary conditions, with a horizontal resolution of $1° \times 1°$ and time resolution of 6 h, used in WRF-Chem were from the global Bias-Corrected Climate Model output, which is the first version of NCAR's CESM [38] (https://rda.ucar.edu/datasets/ds316.1/ (accessed on 20 May 2020)). The daily surface meteorological observation data in NC are supported by the CN05.1 data [39] provided by the China Meteorological Administration (CMA), with a horizontal resolution of $0.25°$(latitude) $\times 0.25°$(longitude).

The anthropogenic emission inventory was an important part of air quality numerical research and prediction. The pollution emission data used in the simulation is based on the China Multi-Resolution Emission Inventory (MEIC) [40], which was developed by the INTEX Inventory team at Tsinghua University and provides a monthly grid emission list with a spatial resolution of $0.25° \times 0.25°$ (http://meicmodel.org/index.html (accessed on 21 May 2020)). The MEIC emissions are representative of the year 2016 with the domain (Figure S2) applicable spans from $30°$ to $45°$N and from $104.8°$ to $125°$E. This inventory was a model of China's anthropogenic emissions of air pollutants and greenhouse gases based on a cloud computing platform. Therefore, it could assess China's carbon emissions more accurately than the IPCC method [41,42]. It covers ten major air pollutants and greenhouse gases including $SO_2$, NOx, CO, NMVOC, $NH_3$, $CO_2$, $PM_{2.5}$, $PM_{10}$, black carbon (BC), and organic carbon (OC) from five sectors such as power, industry, residential, transportation, and agriculture.

*2.2. Data and Methodology*

2.2.1. Selection of Climate Change Scenarios

In this study, we selected the RCP4.5 and RCP8.5 scenarios, which are the two most used by the climate modeling community and represent relatively low and high GHG radiative forcing, respectively. In the RCP4.5 scenario, global emissions of three types of GHGs will peak in 2040; then, GHG (greenhouse gas) concentrations and radiative forcing will stabilize in 2070. Regarding the RCP8.5 scenario, GHG concentrations and radiative forcing will increase over time from 2000 to 2100 [43,44].

2.2.2. Observational Datasets and Model Evaluation Protocol

In order to evaluate the model performance, we used AOD data from satellite retrievals the Multi-angle Imaging Spectroradiometer (MISR) [45] and Medium-Resolution Imaging Spectroradiometer (MODIS) [46] aboard the NASA Terra satellite and Aqua satellite in this study. AOD measurements obtained from satellite remote sensors could provide a cost-effective method as a source of supplementary information for determining the concentration of ground particles. This parameter is proportional to the number of particles in the air and depends on their mass concentration [47]. It could be used to calculate large aerosol content and was the key factor to determine the climate effect of aerosol radiation and the degree of air pollution. So, it could be used as a parameter to estimate

the ground particulate matter [48,49], which is also an important factor to determine the aerosol climate effect.

The model evaluation includes spatial distribution, temporal variation, and statistical analysis. Two sets of satellite data were used to verify the AOD simulated by the WRF-Chem model. For historical periods, based on model validation, we also conducted a basic analysis of the pollutants output by the model during the period 1986–2005. Then, we evaluated PM$_{2.5}$ and BC, sulfate (SO$_4$), and other pollutants closely related to pollutant emissions based on previous studies and conducted quantitative analysis on them. The limitations and uncertainties of model input and the representation of atmospheric processes in WRF-Chem could be better verified by this evaluation and comparison.

## 3. Evaluation of Model Performance

### 3.1. Aerosol Optical Depth

The observed and simulated annual mean AOD in NC in 2000–2005 are shown in Figure 2a–c. Despite there being a slightly bias low at local scales, the observations and modeled results presented similar spatial distribution. There was a large AOD enhancement over industrial and densely populated regions, including the BTH region and the coastal areas. The three data sets all illustrated lower AOD values over north of the study area such as Inner Mongolia and Shanxi Province. In general, compared to MODIS, the simulation underestimated the AOD in NC, which agreed with the previous studies [50]. The reason for the underestimation in NC was the understated injection height of the total dust emissions and biomass burning emissions by WRF-Chem [51,52]. The spatial distribution of AOD could not clearly represent the range and the uncertainty.

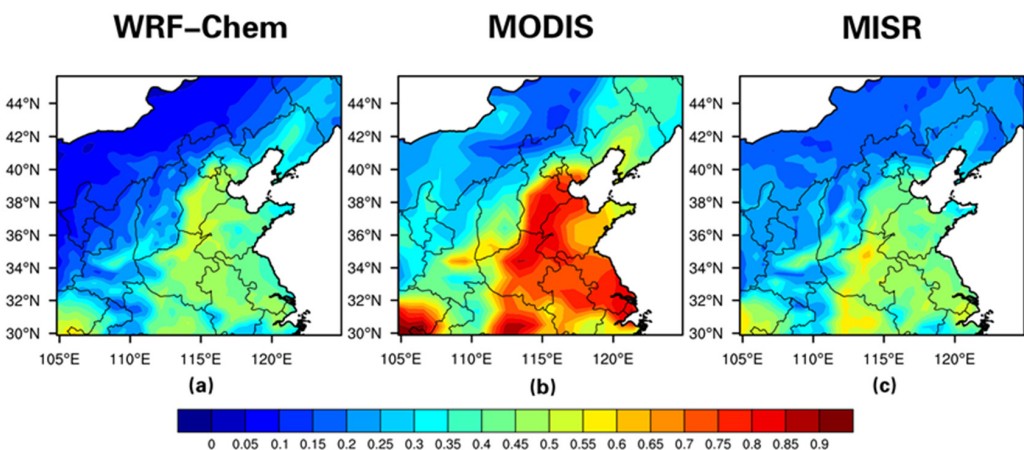

**Figure 2.** Averaged AOD in 2000–2005 derived from model (**a**), MODIS (**b**), and MISR (**c**).

Therefore, we compared simulated annual and monthly time series of AOD during the training period with MODIS and MISR observations. WRF-Chem, MISR, and MODIS data were shown by using the box and whisker plots for average level, degree of volatility, and upper and lower bounds from 2000 to 2005 (Figure 3a,b). The middle line in the boxplot represented the median, bottom, and top lines of the boxplot represent 25th and 75th limits respectively, and the markers at the end of dotted lines represent minimum and maximum values that are not outliers. From Figure 3a, it was found that there was almost no difference between the WRF-chem and MISR for annual variation of median AOD, while some distinction existed for MODIS. The 25th and 75th percentiles of AOD values were close, especially in 2004 and 2005. We also summarized the AOD monthly statistics across different regions in NC (Figure 3b). In spite of a statistically lower number of simulated variations of the monthly AOD (given by the error bar), it was close to the MISR dataset. Column AOD was reasonably well simulated in winter, autumn, and spring, even though there were some discrepancies in summer and April. Compared with MODIS and MISR,

the simulation performed well for median AOD in winter, spring, and autumn. It was interesting to note that the AOD of 2000, 2004, and 2005 was consistent for MISR and the model (Figure 3c); however, underestimations existed between MODIS and the simulation.

**Comparison of simulation and observations**

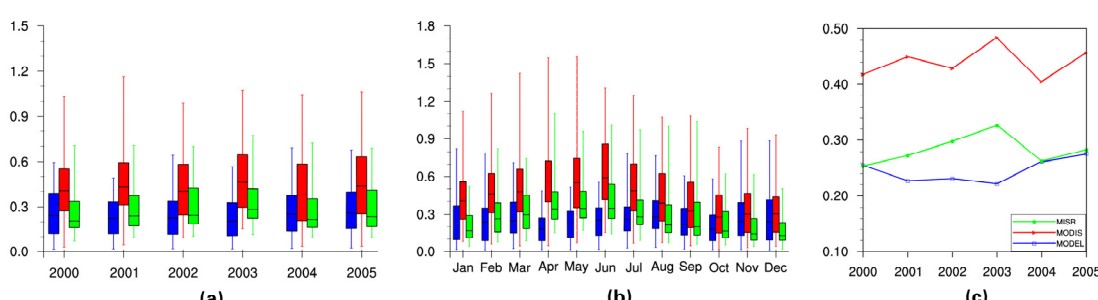

**Figure 3.** (**a**) Individual year averaged with error bar (blue—model, red—MODIS, green—MISR), (**b**) is the same with (**a**) but for monthly average. (**c**) is the year averaged AOD in 2000–2005.

### 3.2. Ground Distribution of Major Components

Figure 4 shows the spatial distribution of annual mean mass concentrations of major chemical components, which include $PM_{2.5}$, BC, OC, $SO_4$, nitrate ($NO_3$), ammonium ($NH_4$), and secondary organic aerosol (SOA) in NC simulated by WRF-Chem from 1986 to 2005. The spatial variation of annual major pollutants concentration is consistent with the spatial distribution of AOD (Figure 2), which also showed the increasing trend of pollutants from northwest to southeast in NC. It showed a high concentration of pollution over the industrial or densely populated regions. $PM_{2.5}$ concentration and its major chemical species showed spatially similar spatial distribution. The maximum annual averaged $PM_{2.5}$ concentration was observed in the mega cities in NC (150 µg·m$^{-3}$), such as Beijing and Tianjin. Similar spatial distributions of other chemical components were also presented. The amount of BC emission in Hebei, Henan, and Shandong provinces was more than those in other places [53].

The annual averaged variation of $PM_{2.5}$ in NC was basically in the range of 26 to 32 µg·m$^{-3}$ (Figure 5). Most of the components of $PM_{2.5}$ were derived from OC, followed by $NO_3$, BC, $NH_4$, SOA, and $SO_4$. The fluctuations in the total $PM_{2.5}$ were influenced by changes of $NO_3$ and OC. Their concentrations were in the range of 5–7 µg·m$^{-3}$ and 12–14 µg·m$^{-3}$, respectively. The annual mean carbonaceous aerosols (the sum of OC and BC) fluctuated during 1986–2005 with a peak in 1995, which was consistent with the results of Streets et al. (2009) [54]. The variations of annual averaged BC, $SO_4$, SOA, and $NH_4$ were relatively stable during 1986–2005. The multi-year averaged concentrations were 4 µg·m$^{-3}$, 1.8 µg·m$^{-3}$, 3.2 µg·m$^{-3}$, and 2.4 µg·m$^{-3}$, while the proportions in $PM_{2.5}$ were about 14%, 6%, 11%, and 9% respectively. The $NH_4$, $NO_3$, and $SO_4$ in atmosphere were basically formed by a gas-to-particle process as a result of chemical reactions of precursor gases [55,56] and were named "secondary inorganic ions". The secondary inorganic ions (the sum of $SO_4$, $NO_3$, and $NH_4$) were in the range of 9 to 12 µg·m$^{-3}$, which comprised one-third of the annual average $PM_{2.5}$ in NC.

The maximum seasonal averaged $PM_{2.5}$ concentrations were observed in winter and the minimum were observed in summer. It was noted that the air pollution in NC mainly occurred in autumn and winter (Figure 5). The highest seasonal average $PM_{2.5}$ that was produced in winter in all urban cities in China may be due to high emissions from the heating period and/or poor dispersion of air pollutants [53]. The OC and BC concentrations in the spring and summer were less than those in the autumn and winter. The seasonal variations of carbonaceous aerosols in winter should be related to the heating activities and biomass burning in this region [57–60].

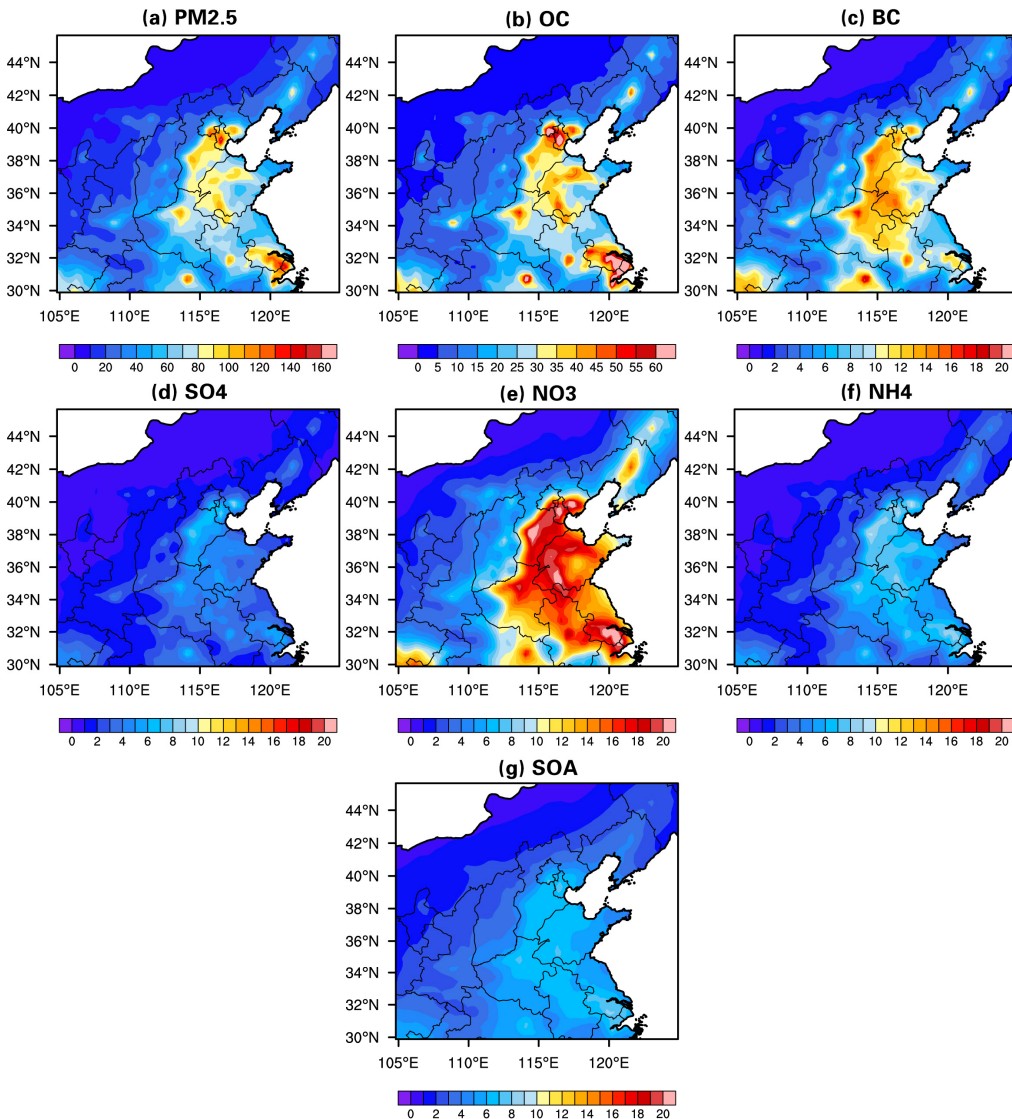

**Figure 4.** The year averaged surface concentration (μg·m$^{-3}$) of (**a**) PM$_{2.5}$, (**b**) BC, (**c**) OC, (**d**) SO$_4$, (**e**) NO$_3$, (**f**) NH$_4$, and (**g**) SOA in 1986–2005.

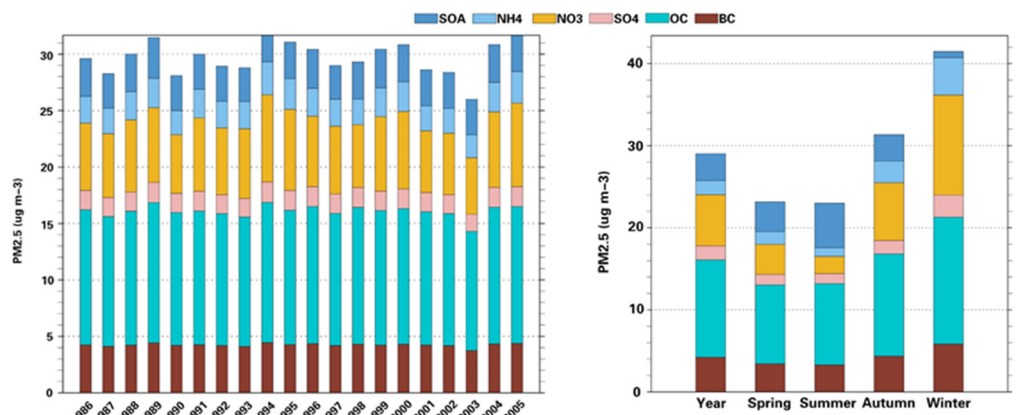

**Figure 5.** The year averaged surface concentration of PM$_{2.5}$, BC, OC, SO$_4$, NO$_3$, NH$_4$, and SOA in 1986–2005.

More coal consumption combined with a stable meteorological condition might be the reason for the high PM$_{2.5}$ concentrations in winter [61–64]. The secondary inorganic ions also have obvious seasonal changes; e.g., secondary inorganic ions accounted for 50% of PM$_{2.5}$ concentration in winter and only 18% in summer. It was noted that NO$_3$ increased obviously from 8 to 12 µg·m$^{-3}$, which was one of the main reasons for the increase in PM$_{2.5}$ concentration in winter. Based on consistency with previous studies, it proved that the model reproduced the air pollution distribution in the historical period in NC.

## 4. Projection of Air Pollution and Meteorological Conditions in the RCPs Scenarios

### 4.1. Projection of Air Pollution

Figure 6 shows the spatial distribution of the seasonal averaged AOD in the RCP4.5, RCP8.5, and their changes with respect to historical period. The seasonal average AOD in summer (JJA, June–July–August) was the greatest, followed by winter (DJF, December–January–February), autumn (SON, September–October–November), and spring (MAM, March–April–May).

In the two RCPs, their mean AOD was in the range of 0.08–0.48 in spring, 0.16–0.68 in summer, 0.12–0.48 in autumn, and 0.08–0.56 in winter, respectively. The seasonal variation of AOD had the peak value of 0.68 in summer over BTH. One of the most important factors was the highest PBLH in summer. Secondly, the summer temperature and humidity were high, which were conducive to the formation of aerosols [65]. The greatest values in other seasons were 0.52, 0.48, and 0.44 in winter, autumn, and spring respectively, which were located in the areas close to mega cities, e.g., Beijing and Tianjin. Compared to the historic period, the junction of Inner Mongolia and Liaoning had a slightly larger increase than other regions in the summer under two scenarios. The AOD showed the largest reductions with the range of 0.02–0.08 in the RCP4.5 scenario over NC in autumn, especially in Hebei and Henan Province. The spatial distributions of projected AOD under RCP8.5 were similar to those under RCP4.5. It was noted that in winter, the changes of simulated AOD basically demonstrated different spatial performance under two scenarios. AOD was projected to decrease in the study region of NC excepted to BTH. However, the simulated AOD performed an obvious increase in the whole NC in the RCP8.5 scenario, with a maximum increase exceeding 0.06. This may be attributed to the decrease in precipitation in winter, but high humidity is conducive to the formation of aerosols. We further compared the seasonal differences of AOD between the RCP4.5 and RCP8.5 scenarios (bottom panel in Figure 6). In RCP4.5, AOD was slightly less than that under the RCP8.5 in NC in spring and summer, with a value of 0.02–0.04. Compared with RCP4.5, AOD in autumn showed a minor increase in western NC but decreased in BTH under the RCP8.5. Importantly, in winter, RCP8.5 could lead to an apparent increase of AOD in NC against RCP4.5, such as in BTH and Henan province. That meant more serious pollution in winter in the RCP8.5 compared to that in the RCP4.5.

Figures 6 and 7 show the spatial distribution of PM$_{2.5}$ in the two RCP scenarios and their differences. It was found that the spatial distributions of PM$_{2.5}$ concentration and AOD were similar in the two RCP scenarios. The PM$_{2.5}$ concentrations both demonstrated the maximum in winter and the minimum in summer. This may be attributable to more frequent rainfall in summer, which could remove PM$_{2.5}$ [65]. The averaged PM$_{2.5}$ concentration in most of the areas of NC were in the range of 10–100 µg·m$^{-3}$ in spring and summer. In autumn, there was a concentration of PM$_{2.5}$ 100–200 µg·m$^{-3}$ in most regions. The averaged concentration was 100–250 µg·m$^{-3}$ in winter. This was due to the increased emission derived from the heating winter, low PBL, and less precipitation [66,67]. The simulated PM2.5 concentrations in NC were consistent with previous studies and corresponding to the frequent occurrence of haze pollution in recent years [12,68].

In consideration of the difference between historic period and the future, PM$_{2.5}$ levels in the RCP4.5 are expected to be controlled well except for autumn PM$_{2.5}$ concentrations in BREC with an increase range of 4–12 µg·m$^{-3}$. In winter, the largest reduction of PM$_{2.5}$ concentrations with the range of 10–12 µg·m$^{-3}$ were in Tianjin, Hebei, and Shandong

provinces. Jiang et al. (2013) also showed the largest decreases in winter in NC, which was based on GEOS–Chem models [69]. However, the degree of reduction was slightly different. The large reduction in PM$_{2.5}$ concentration in our simulation could be attributed to the difference in emission reduction between the RCP and A1B scenarios and our higher model resolution, which can better represent the oxidant environment and transportation process. In the RCP8.5, the seasonal spatial distribution changes in spring and summer were slightly similar to those in the RCP4.5. In autumn, there was an apparent reduction in central Hebei, with values of 4–10 $\mu g \cdot m^{-3}$. Furthermore, PM$_{2.5}$ increased by about 6–10 $\mu g \cdot m^{-3}$ in the coastal part of Hebei and Shanghai province. In winter, there was a large increase mainly in the central area of NC, with values of 2–8 $\mu g \cdot m^{-3}$.

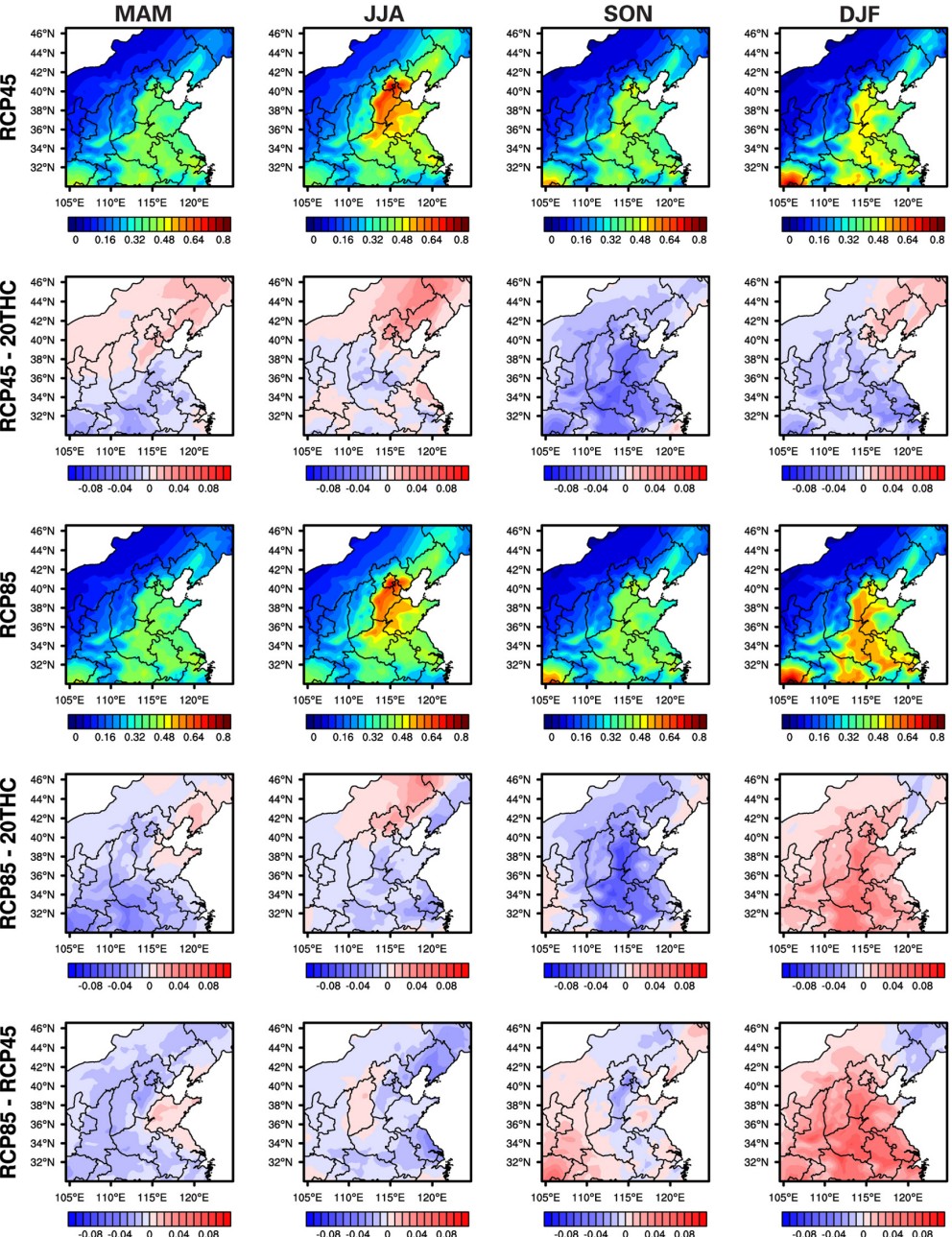

**Figure 6.** Averaged AOD in MAM, JJA, SON, and DJF in the RCP4.5 and RCP8.5 scenarios, and their changes relative to historical period. Figures in the bottom are represented the differences of AOD between the RCP8.5 and RCP4.5 scenarios.

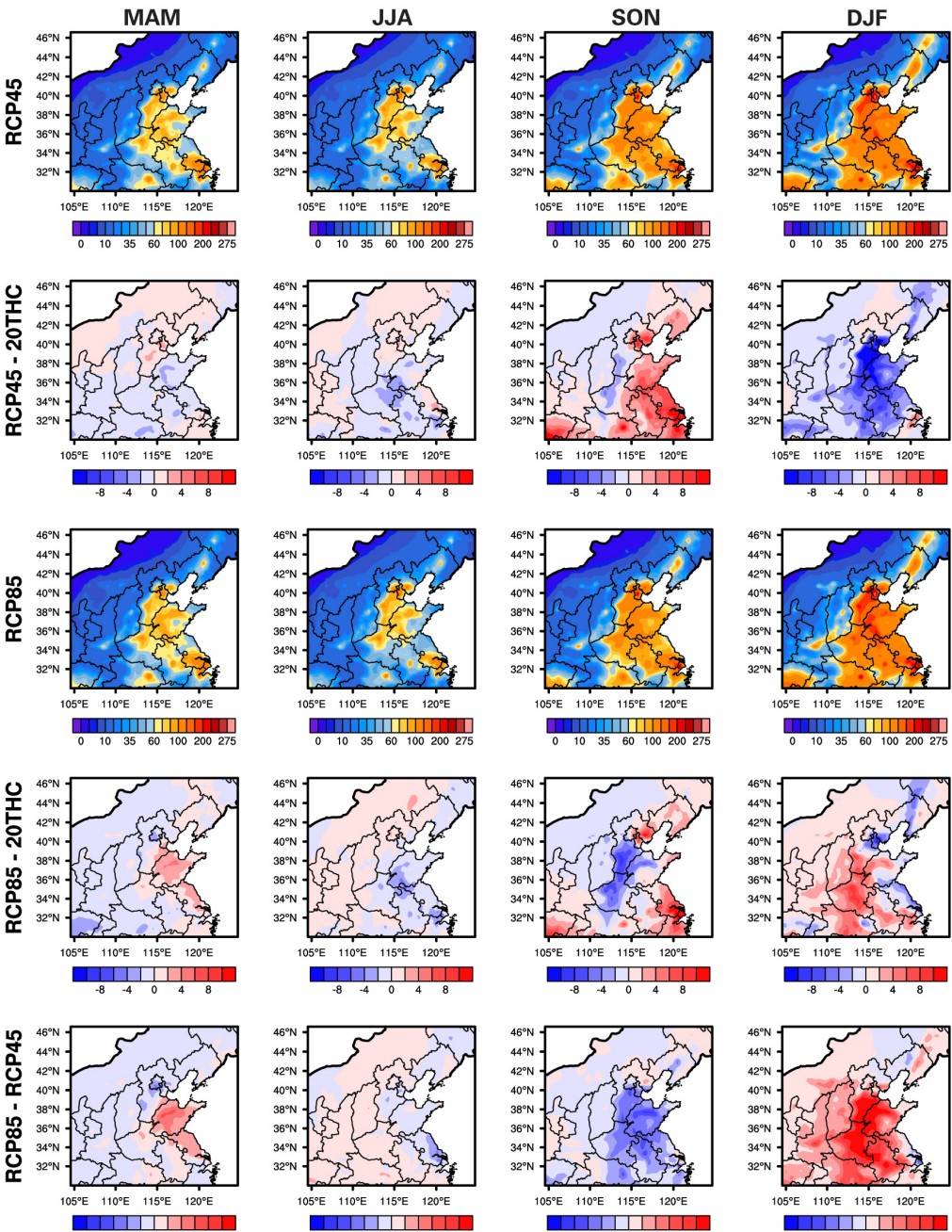

**Figure 7.** The same as Figure 6 but for the surface concentration of PM2.5.

There were also some points worth noting about the difference between the two scenarios. Compared to RCP4.5, the $PM_{2.5}$ concentration in the RCP8.5 scenario at Shandong province increased by 2–8 $\mu g \cdot m^{-3}$, whereas it decreased by 0–4 $\mu g \cdot m^{-3}$ in BTH in spring. In autumn, the concentration of $PM_{2.5}$ decreased in most of the regions of NC, especially in Shandong in southern Hebei province, with values of 4–10 $\mu g \cdot m^{-3}$. In winter, the comparison results revealed more $PM_{2.5}$ concentrations in NC under the RCP8.5 scenario, with the increase of 10–12 $\mu g \cdot m^{-3}$. Gao et al. (2021) also showed the largest increase of $PM_{2.5}$ concentration in the RCP8.5 scenario over the NC region based on RAMS-CMAQ during 2045–2050 [17]. The predicted changes in the annual average concentration of $PM_{2.5}$ and AOD under RCP4.5 and RCP8.5 provided possible options for the control and reduction of $PM_{2.5}$ air pollution nationwide.

### 4.2. Projection of Meteorological Conditions

The previous study indicated that the meteorological elements, e.g., precipitation and boundary layer height, were attributed to the interdecadal changes (or trends) of regional air pollution in China [70]. For example, precipitation had a removing effect on atmospheric pollution. PBLH was related to vertical mixing, which affects the dilution of pollutants emitted near the ground through various interactions and feedback mechanisms. In this section, we explored the seasonal changes of precipitation (Figure 8) and PBLH (Figure 9) caused by the warming in the future and tried to find the connection between meteorological elements and air pollutions.

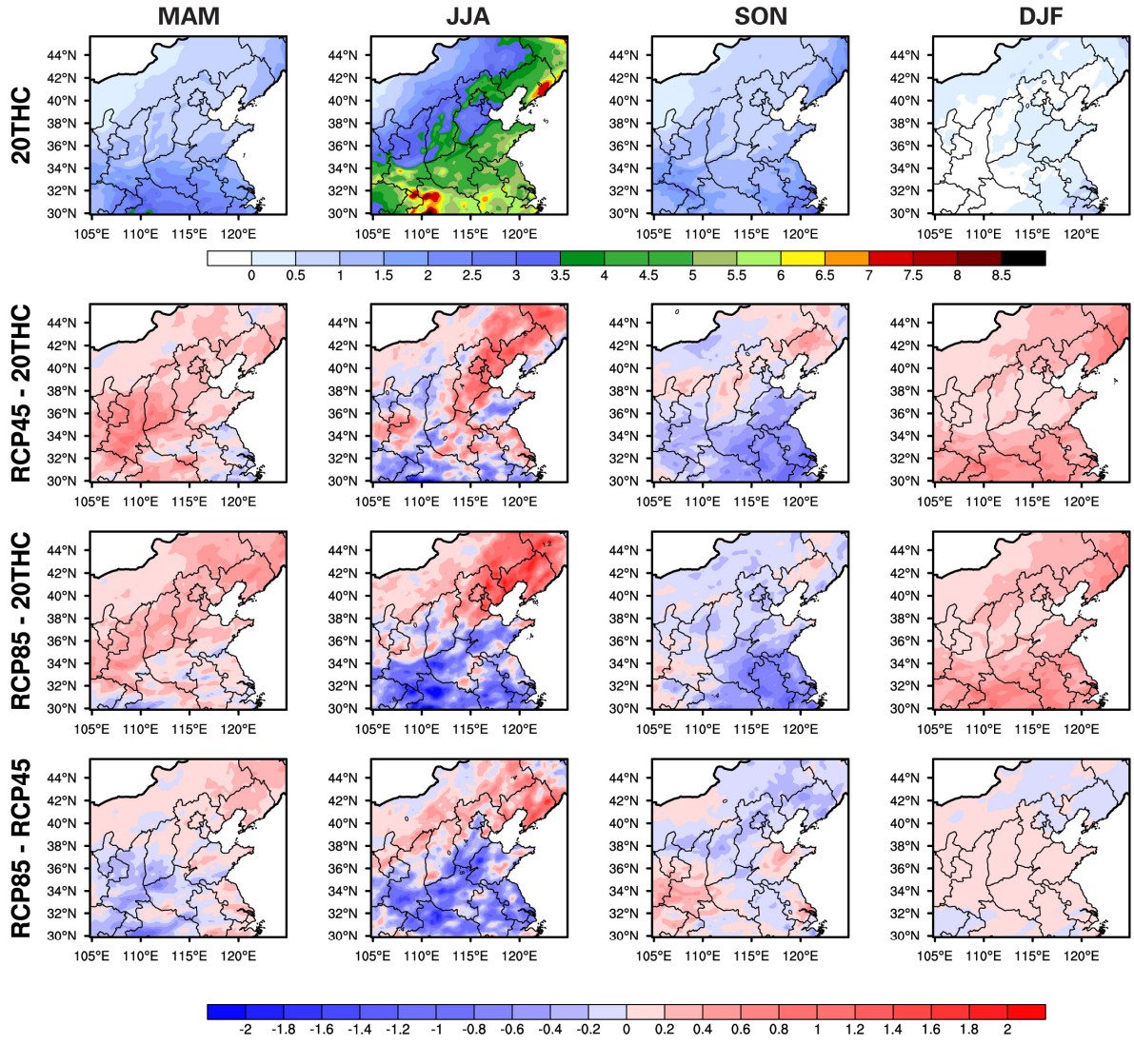

**Figure 8.** Projection of precipitation in MAM, JJA, SON, and DJF. Changes in precipitation which are represented the RCP4.5 minus 20THC, RCP8.5 minus 20THC, and RCP8.5 minus RCP4.5.

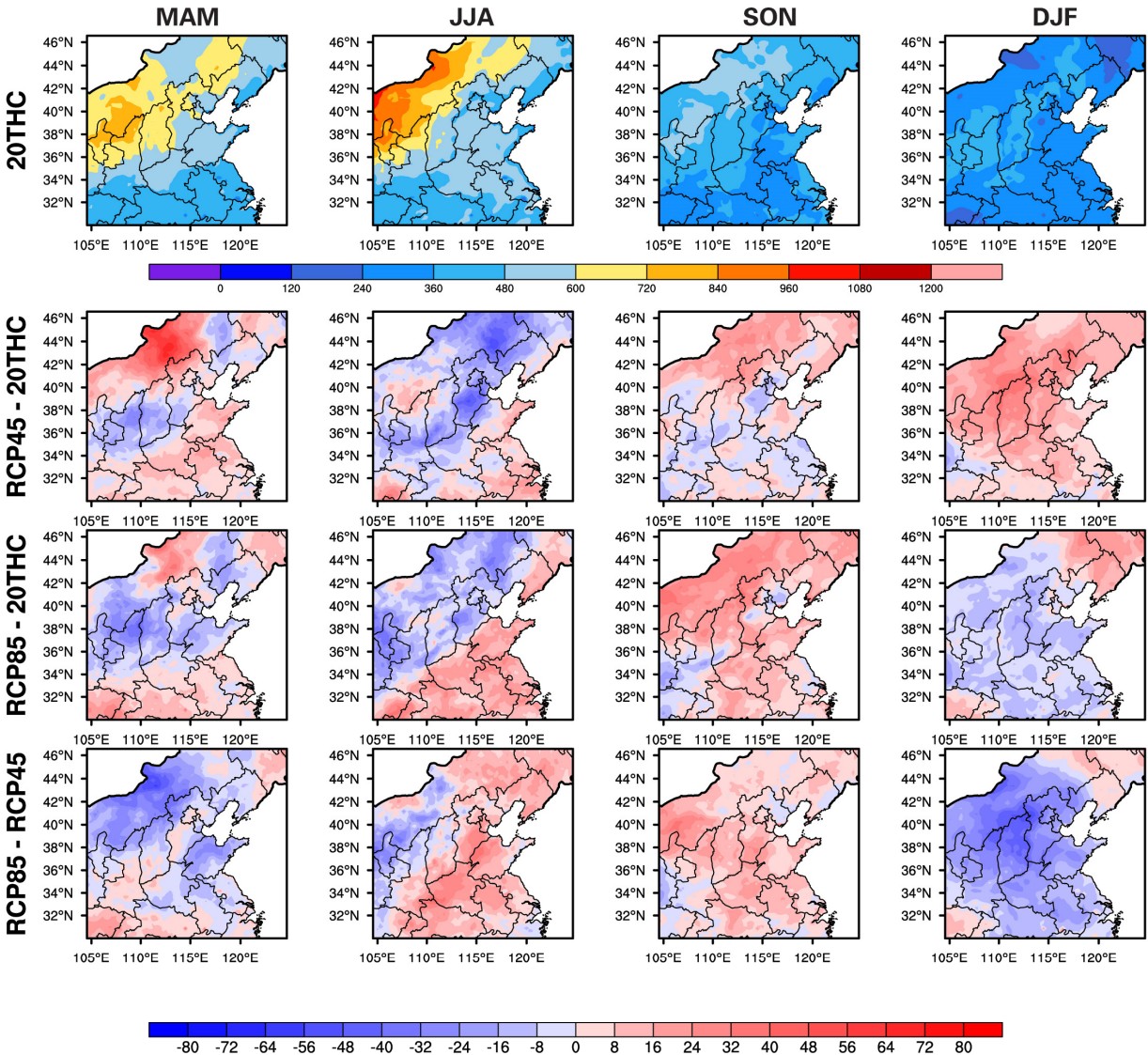

**Figure 9.** The same as Figure 8 but for the level of boundary layer.

Precipitation in NC was mainly concentrated in summer due to the influence of the East Asian monsoon, followed by autumn, spring, and winter. Compared to the historic period, precipitation was projected to increase over all of NC in spring under the two scenarios. It is noted that the greatest increases of precipitation were in summer, with a value of 1.5 mm/day in BTH. On a regional scale, the changes in precipitation responded relatively well to the pollution (Figures 6 and 7) in spring and summer. In autumn, changes of precipitation both showed entire decreasing patterns. Under RCP4.5 and RCP8.5 scenarios, the overall precipitation increased in the entire region of NC in winter, with values of 0.2–0.6 mm/day. The increase in winter pollutants under the RCP8.5 scenario (Figure 7) may be due to the insignificant increase in precipitation, but it provides humidity conditions for air aerosols that are conducive to aerosol moisture absorption growth, thereby increasing the concentration of $PM_{2.5}$ [66].

A comparison of within-scenario results revealed that changes in the summer were similar to those in spring, but the values were greater. There was decreased precipitation in the range of 0.6–1.2 mm/day in southern Hebei and its surrounding regions. The result

showed that the increase in relative humidity in winter under the RCP8.5 scenario caused more serious pollution

The projected PBLH under the two RCP scenarios and their difference are shown in Figure 9. The PBL is highest in summer and lowest in winter. The possible explanation is that the higher solar radiation and heat flux in summer lead to stronger surface heating, which in turn produces stronger turbulence and convection [71]. Compared to the historic period, PBLH increased in most of NC except for the central region in spring under two scenarios. In the summer, it is observed that most of the land areas of the NC domain receive lower PBLH, especially in BTH, with decrement of 8–40 m. While in autumn, a large increment of PBLH was found in basically the entire NC region. Correspondingly, both the $PM_{2.5}$ concentration and AOD in autumn decreased (Figures 6 and 7). In winter, PBLH had undergone opposite changes under the two scenarios. It was mainly due to the variations of turbulence processes induced by the different thermal states between the two scenarios. RCP8.5 caused a decrease, which directly affected the distributions and magnitudes of PM2.5 concentrations in NC along with PBLH changes. The difference between the two scenarios revealed that PBLH decreased by 16–31 m in spring. However, in summer and autumn, the PBLH increased in the RCP8.5 scenario compared with the RCP4.5 scenario, with a range of 8–32 m. RCP8.5 could cause an obvious decrease in PBLH in winter, with the maximum reduction of 48 m in southern Hebei and its surrounding areas. It was found that the lower the local PBLH, the lower the turbulence intensity, which is not conducive to the diffusion of pollutants.

## 5. Conclusions and Discussion

In this paper, WRF-Chem driven by the MEIC inventory was used to simulate the seasonal changes of air pollution in NC in the RCP4.5 and RCP8.5 scenarios. The results of the evaluations showed that WRF-Chem could capture the spatial distributions variations of AOD at annual and seasonal temporal scales.

In the RCP4.5 and RCP8.5 scenarios, the projected spatial distributions of AOD were similar, with the maximum in summer. The greatest concentration was located in BTH, which averaged AOD peak values of 0.44, 0.68, 0.48, 0.52 across four seasons, respectively. The maximum $PM_{2.5}$ concentration was in winter, followed by autumn, summer, and the lowest in spring, with values of 10–250 $\mu g \cdot m^{-3}$, 10–200 $\mu g \cdot m^{-3}$, 10–100 $\mu g \cdot m^{-3}$, and 10–100 $\mu g \cdot m^{-3}$, respectively. AOD and $PM_{2.5}$ showed the great variation in autumn and winter but weaker changes in spring and summer. In the two RCP scenarios, AOD showed the maximum reduction in autumn compared with other seasons in BREC. In the RCP8.5, AOD was effectively increased. A comparison between the changes in two scenarios revealed that increased AOD was presented in the RCP8.5 pathway.

In the RCP4.5, the maximum decreased $PM_{2.5}$ in the range of 2–12 $\mu g \cdot m^{-3}$ occurred in winter. Compared to RCP4.5, there was a higher decline in autumn for RCP8.5, especially in the coastal region, with values of 4–10 $\mu g \cdot m^{-3}$. However, there was more $PM_{2.5}$ concentration in the range of 4–12 $\mu g \cdot m^{-3}$ in winter for RCP8.5 compared to RCP4.5, which indicated heavy pollution while selecting a high emission pathway.

Meteorological conditions also had a large connection with the projection of air pollution. Actually, the impact of meteorological conditions on air quality was not a single one but rather the result of a combination of multiple factors. In autumn, precipitation decreased, but PBLH increased in the two RCPs, which partly explained the decrease in $PM_{2.5}$ concentration and AOD in Hebei and the surrounding areas. In winter, the reduction of $PM_{2.5}$ and AOD in BERC could be affected by the increased precipitation and PBLH in the RCP4.5. A high GHGs emission pathway induced decreased PBLH, which would be further related to the increased air pollution through vertical mixing.

In summary, in the RCP8.5 scenario, the severe air pollution in NC was expected to continue, especially in winter, but the situation was alleviated in the RCP4.5 scenario. Our research provided a meaningful assessment of future emission scenarios. The RCP4.5 pathway probably improved air quality compared to that in the RCP8.5 scenario. Actually,

the projection of air quality depended on the assumptions of future climate status and emission scenarios, which had great uncertainties. Nevertheless, this study assessed the impact of different scenarios on air pollution, which would help to guide the development of emission control strategies.

**Supplementary Materials:** The following are available online at https://www.mdpi.com/article/10.3390/rs13163064/s1, Figure S1: The structure of WRF-Chem (derived from WRF-Chem Version 3.9.1.1 User's Guide), Figure S2: The emission distribution in China of MEIC Inventory in 2016.

**Author Contributions:** C.D.: Validation, Formal analysis, Writing—Original Draft, Visualization. Z.J.: Conceptualization, Methodology, Investigation, Writing—Original Draft, Writing—Review and Editing, Supervision. Y.X.: Software, Formal analysis. Z.H.: Validation. X.Z.: Resources, Formal Analysis. W.D.: Supervision. All authors have read and agreed to the published version of the manuscript.

**Funding:** This study was supported by the key National Natural Science Foundation of China (91644225), the National Key Research and Development Program of China (2016YFA0602701), Innovation Group Project of Southern Marine Science and Engineering Guangdong Laboratory (Zhuhai) (No. 311020008), and the Open Program (SKLCS-OP-2020-9) from the State Key Laboratory of Cryospheric Science, Northwest Institute of Eco-Environment and Resources, Chinese Academy of Sciences.

**Institutional Review Board Statement:** Not applicable.

**Informed Consent Statement:** Not applicable.

**Data Availability Statement:** The simulation data are available on request from the corresponding author.

**Conflicts of Interest:** The authors declare no conflict of interest.

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
