# Peer review of "Projection of Air Pollution in Northern China in the Two RCPs Scenarios"

_remotesensing, doi:10.3390/rs13163064_

Round 1
Reviewer 1 Report
The influence of climate change on air pollutant concentrations is worthy to investigate, especially in view of future development world economy and urbanization. Due to spatial and temporal variability of meteorological fields under different climate change scenarios, the modelling is a useful tool to predict air pollutants concentration fields in the future. The study uses the Weather Research and Forecasting Model with Chemistry(WRF-Chem) to investigate the changes in PM2.5 concentrations and Aerosol optical depth (AOD) over North China under two future climate change scenarios. The findings, namely that changes of meteorological conditions have great impact on air pollution are not of surprise but the evaluation of the changes relative to historical period (1986-2005) might give city planners a relevant tool into their hands.
On the one hand, I believe that the authors have collected an interesting and extensive database and have carried out the proper analyses. On the other hand, I found some of the description of the paper unclear or even missing. Therefore, I recommend that a revision is warranted.
General comments
The authors actually raised four issues: averaged (in the period 1986-2005) ground distributions of major PM2.5 components; evaluation of the model performance on the example of AOD using satellite data (from the period 2000-2005); projection of AOD and PM2.5 concentrations in different regions in North China under two future climate change scenarios (for the period 2031-2050); projection of changes precipitation and boundary layer height under these scenarios, while according to the title of the work, they should focus on the issue: Projection of air pollution in NC under two future climate change scenarios.
Therefore, first of all, they should present these two scenarios, and the baseline scenario, briefly discuss how the meteorological conditions are changing (these scenarios include changes in precipitation and the height of the boundary layer) and the emission source data. The emissions distribution in China under different scenarios should be shown. At present, there is only information that an emission inventory was available for 2012, but how this issue relates to years 1986-2005 and 2031-2050. The model structure should be in the supplementary file.
Specific comments
The abstract presents the scope of the research and lacks a purpose. After specifying the purpose of the research, it will be easier to formulate the most important conclusions. They are currently blurry. For example, this issue has not been considered: l. 22-23: Results indicated that air pollution would be reduced in winter if society developed in the low emission pathway.
- 83-89 - This justification for undertaking the research does not convince me. Previously, the authors concluded that there are similar studies in the literature (l.61-69).
Why was the model evaluation carried out for years other than the baseline scenario?
Fig. 4 - units are missing
- 234 –It should be : Figure 5b.
Figs 6-8 - please explain the symbols MAM,JJA, SON, DJF
Reviewer 2 Report
Please see the attached PDF file.

Reviewer 3 Report
Dou et al., used the WRF-Chem model as a regional climate to predict the response of AOD and concentration of PM2.5 with respect to climate change in future over Northern China. They compared the results by applying the two RCPs scenarios. Prior to this, as a first step, they evaluated the model performance for the simulated AOD against satellite datasets of MODIS and MISR. Overall, I found the subject interesting, because the authors tried to predict aerosol concentrations in the future over east Asia that is thought-provoking question. And because of using satellite retrievals, the study is likely to be of interest to the “Remote sensing” readership. However, the general idea and the applied method were not original. The results are not presented in a clear manner and the manuscript is not well written. I suggest publication of the work in “remote sensing”, after the major points listed below are addressed:
General comments:
First of all, the manuscript gives the impression of being rushed. There are lots of grammatical and typo errors that should have been picked up, few samples:
In several places there is inconsistent verb tense, for example “The horizontal resolution is 27 km and the vertical grid in the model consists of 27 levels ranging. The Lin microphysics scheme [32] was used in this simulation. …“ on page 3 line 103-104.
Line 15: aerosols’ concentration aerosol concentrations
Line 15-17: Future Changes of air pollution in the middle of 21st century (2031-2050) was projected in the two Representative Concentration Pathways (RCP4.5 and RCP8.5) against to the situation in the historical period (1986-2005). it is incoherent, should be rephrased
Line 20: a average an average
Line 23: was projected to increasing increase
Line 217: SOA, SO4 SOA and SO4
Line 234: (Figure. 4b) (Figure 5b)
Line 251: histirical historical
Line 323: PBLH PBLH (the planetary boundary layer height). The abbreviation should be described before applying
Line 378: The great concentraion concentration
Line 378-379: The great concentraion was located in BTH, which averaged AOD peak values was 0.44, 0.68, 0.48, 0.52, respectively it is incoherent. respect to what???
Lines 391 : emssion emission
Line 398-400: “A high GHGs emssion pathway induced to decreased PBLH that would further had connection with inceased air pollution through vertical mixing.” emission, decrease, have further, increased
and much more!!!!
The authors should also reconsider the information state in the Introduction. The author just put some inconsistent and non-relevant text together. Furthermore, they reference studies that have compared different scenarios in their studies, but without any information about the relevance of those studies to their comparison and evaluation. For example, on page 2 line 63-67, the authors state that “Yahya et al. (2017) used WRF-Chem to project the climate and air quality in the U.S from 2046 to 2055 in the RCP4.5 and RCP8.5 scenarios [16]. Gao et al. (2021) applied CESM (NCAR's Community Earth System Model) to analyze the differences in aerosol concentrations and annual variations in the RCP2.6, RCP4.5 and RCP8.5 66 scenarios, respectively [17] . . .”, but do not provide information about the major findings of these studies or at least the information from these studies that is relevant to their own.
I highly recommend that the author also rewrite the result sections, especially section 4. The authors pointlessly extract just lots of unimportant and redundant number and statics that are easily can be reached out at the figures as well, in many paragraphs. They do not provide an appreciated explanation for any single result. To justify a publication, it would be necessary to concentrate on few new outcomes, which has not yet been done consequently. Therefore, I recommend revising the manuscript in order to highlight and analyze main valuable results that have so far not discussed in previous publications together with interpretation of the outcome.
General questions:
In Figures 6 and 7, differences of the predicted AOD and concentration of PM2.5 whether between the two simulations using RCP8.5 and RCP4.5 scenarios (the fifth row) or simulations for the future scenarios and historical period (the second and furth rows) are absolutely small against the average predicted AOD and concentration of PM2.5 (the first or third rows). How can the author conclude whether two model simulations are significantly different from one another? In other words, are the difference in mean value of the two data set statistically significant in comparison with the variance between the data in two series from their respective means? I highly encourage the author, for making their results reliable, to perform a statistical practice using Two-tailed t-test (inspired by Hedegaard et al., 2008- https://doi.org/10.5194/acp-8-3337-2008).
Can the author explain why the model predict the planetary boundary layer height different for two scenarios (Figure 9, row3)?
Round 2
Reviewer 2 Report
The authors did a good job improving the manuscript. However, I feel that the manuscript needs a language check or proofreading. At some parts, the manuscript is too wordy and could be shortened.
Reviewer 3 Report
The authors have tried to address the comments raised and incorporated them in the revised manuscript, especially with clarification and discussion of the results. Overall, I think the manuscript merits publication and could be useful for the rest of the scientific community.